# Tensile Damage Study of Wind Turbine Tower Material Q345 Based on an Acoustic Emission Method

**DOI:** 10.3390/ma14092120

**Published:** 2021-04-22

**Authors:** Xiao Tang, Lida Liao, Bin Huang, Cong Li

**Affiliations:** 1School of Energy and Power Engineering, Changsha University of Science and Technology, Changsha 410114, China; tangxiao@stu.csust.edu.cn (X.T.); liconghntu@csust.edu.cn (C.L.); 2UniSA STEM, University of South Australia, Mawson Lakes, Adelaide, SA 5095, Australia; bin.huang@unisa.edu.au

**Keywords:** acoustic emission technology, wind turbine towers, normalized cumulative parameters, damage analysis

## Abstract

As essential load-bearing equipment to support the nacelle and blades, the tower is subjected to the whole wind turbine loading. This study proposes a new method of combining acoustic emission and normalized accumulation parameters to characterize wind turbine towers Q345 steel damage. First of all, tendency analysis of the acoustic emission signal parameter was conducted to determine damage degree during the damage stage. Secondly, we normalized the accumulation of amplitude and other parameters to compare the proportion of each accumulation parameter at different stages, while studying the spectra of common acoustic emission signals. Finally, comparing the differences and similarities of the normalized accumulation parameters between three different rates, we analyze the effect of rate on the normalized accumulation parameters. These results indicate that the normalized cumulative duration parameter is suitable for characterizing the yield damage occurrence, the normalized cumulative energy parameter is very sensitive to the fracture stage, the normalized cumulative energy parameter is least influenced by the loading rate, and the energy parameter is a sensitivity factor for normalized expression, which to realizes the stage of damage judgment.

## 1. Introduction

According to the Global Wind Energy Council report [1], wind energy has become a cost-competitive clean energy source in the global mainstream over the past 20 years. Wind turbines are the crucial equipment for wind energy to be converted into electricity. In terms of wind turbines’ material properties, the tower requires high strength and stiffness to ensure proper operation of the wind turbine [2]. However, the collapse of wind turbine towers has not stopped, and in September 2019, a GE wind turbine collapsed at Delta 6 wind farm in Maranhão, Brazil. The accident was the result of failure to detect irreversible plastic damage inside the material of the tower and the bolts to which it was attached in time, and a force imbalance caused the tower to pull off. This has created a huge risk to people’s safety. As can be seen, one of the important works is the investigation of tensile damage in wind turbine tower materials. This study of tensile damage is a critical way to evaluate the load-bearing capacity of the material.

The acoustic emission (AE) technique is a promising non-destructive surveillance technology for materials study and in situ supervision of structures [3]. The acoustic emission detection technique has been widely used in damage detection of metallic materials [4,5,6], composite materials [7,8,9], and rock materials [10,11,12,13]. Ennaceur et al. [6] studied the damage and cracking of two metallic materials, 304L and P265GH steel, the using acoustic emission technique, which found that acoustic emission parameters such as count and amplitude can be used to characterize different crack propagation mechanisms. Saeedifar et al. [9] investigated the damage characterization of composites in two aspects; the first part is aspect is the diagnosis of damage and the second part is the strength prediction of composites by combining acoustic emission techniques and three different models. Kourkoulis et al. [10] combined acoustic emission technology, digital images, etc. to monitor the damage extension process in Dionysos marble. The results show that the cause of damage in Dionysos marble can be known from the signal perspective.

Acoustic emission detection techniques have been applied to damage detection of wind turbines, including damage detection of wind turbine composite blades [14,15], damage detection of wind turbine gearboxes [16,17], and damage detection of wind turbine bearings [18]. Tang et al. [19] proposed a pattern recognition method for classifying different damage mechanisms of wind turbine blades from fatigue tests. Gomez et al. [20] successfully detected fiber fracture damage in wind turbine blades and accurately identified the defect locations. Li et al. [21] used a damage classification method based on acoustic emission technology for gearbox fault classification and the results showed that its classification rate was higher than other methods. The acoustic emission technique is relatively stable in terms of fault detection performance and damage class separation capability. We can obtain material damage fault information in advance from the signal perspective for the purpose of improving material performance and material fault early warning.

After understanding the acoustic emission characteristics of the damage process, people have been made to quantify the material damage process through the acoustic emission signal. The first step in the analysis of material damage using acoustic emission techniques is to correlate the acoustic emission parameters with the damage process and then characterize the progress of the damage by means of acoustic emission parameters. The AE parameter signal originates from the object to be examined, so the examination progress does not influence the regular operation of the equipment. There is a one-to-one correlation between the obtained acoustic emission signals and the dislocation activity at various stages of distortion [22]. Therefore, the proper processing of the obtained acoustic emission signals allows the correct identification of the degree of deformation damage of the metal [23]. The methods for processing the acoustic emission signal characteristics of the damage process are parametric and waveform analysis approaches. The parametric analysis approach analyzes the endomorphism of the damage source based on the trend of the parameters characterizing the waveform during the damage process and the correlation or distribution between the parameters. The representative AE signal contains a number of parameters, the common ones being amplitude and counts. In early studies, researchers found that the commonly used acoustic emission parameters (amplitude, count) can be used to connect the signal to the damage process. Nair et al. [24] found that the accumulated acoustic emission data under continuous loading can be used as a structural damage assessment to continuously monitor the damage of the structure condition. Lugo et al. [25] developed a model to quantify the damage of 7075 aluminum alloy under tensile loading by correlating the acoustic emission counts with the number density and demonstrated that the acoustic emission activity was directly related to the damage progression of this alloy. However, with further research, it was found that commonly used acoustic emission parameters such as amplitude are susceptible to signal attenuation caused by threshold settings and the external environment, so new parameters were continuously searched for to accurately describe the signal–damage relevance. Fernando et al. [26] proposed to introduce instantaneous amplitude, wavelet analysis, etc. to solve the defects of threshold detecting AE hits, which found that the Akaike information criterion and continuous wavelet transform could enhance onset measurements. Babua et al. [27] proposed a new characteristic parameter, peak amplitude of acoustic emission impact counts, to correlate the fatigue damage process with the characteristic parameter to distinguish different damage stages. Chai et al. [28] proposed to measure the damage of a material by acoustic emission entropy, independently of the threshold and other temporal driving parameters. Compared with the acoustic emission amplitude parameter, the new parameter is more useful in identifying various damage stages and critical damage. At the same time, studies on damage models and acoustic emission parameters have been developed. Yu et al. [29] developed models for the absolute acoustic emission energy rate, strength of stress, fracture toughness and load ratio to forecast the residual life at different damage stages. Barat et al. [30] monitored the low-temperature tensile deformation of AISI type 304 stainless steel and proposed from cumulative acoustic emission counts in low-temperature tests a mechanistic model of martensite evolution. Gagar et al. [31] developed a novel method for fatigue crack length estimation by correlating the applied cyclic load with the acoustic emission signal generated during crack extension. Using the average of the normalized absolute errors, predictions in the range of 0.28 to 0.4 were obtained. Keshtgar et al. [32] found that the linear model used to correlate the acoustic emission parameters and crack extension was not related to the loading conditions and loading frequency and developed a correlation model between acoustic emission parameters and fatigue cracks. Barsoum et al. [33] proposed acoustic emission technique and neural network for A572 steel to identify damage and fatigue life prediction, and created a backpropagation neural network fatigue life prediction model to provide ideas for the life study of structural steel parts. Although these studies provide research ideas for dealing with acoustic emission parameters and material damage, these studies do not cumulatively normalize the acoustic emission parameters and thus characterize the different damage stages of the material.

In summary, despite a large number of studies dedicated to the enhancement of various material properties of wind turbines and early warning of material failures by acoustic emission methods, there is still a gap in accurately and objectively analyzing the changes of signal parameters during the damage process. In this paper, a method combining acoustic emission signal parameters with normalized accumulation is proposed that can be used to detect wind turbine tower material damage by normalizing the accumulation of amplitude, ringing count, energy, impact and duration parameters during the damage process to compare the proportion of each accumulation parameter among different stages, so as to obtain the changes of different accumulation parameters in different damage stages to achieve the goal of characterizing the damage evolution of the material. In this paper, the signals of Q345 steel tower material during tensile damage were investigated using the acoustic emission technique and normalization method, which can help to determine the damage stage of the material and characterize the damage process. The rest of this paper is structured as described below. Section 2 describes the experimental materials, methods and preparation before the experiment. Section 3 discusses the changes of the signal parameters during the damage process by performing spectral analysis and normalization of the signal parameters. The similarities and differences of the normalized cumulative parameters and hits are compared by different rates. Finally, a summary and suggestions for future research are presented in Section 4.

## 2. Materials and Methods

### 2.1. Materials and Specimen

The specimen material is Q345 steel for wind turbine tower construction, and its chemical composition is shown in Table 1. The specimens are prepared according to GB/T228-2008 standard, the specimen size is shown in Figure 1, the length is 134 mm, the width is 22 mm, the thickness is 4 mm. the number of specimens is 15 in total, the specimens are divided into 5 groups, each group having 3 specimens. The first, second and third groups of specimens were tested under loading rates of 1 mm/min, 1.5 mm/min and 2 mm/min, respectively, as a comparison of the loading rate conditions. The fourth and fifth sets of tests were performed at a loading rate of 1.5 mm/min, and together with the results of the second set of tests were used as data for subsequent damage quantification.

### 2.2. Tensile Tests

The machine used in this tensile test is the WDW-300E electronic universal testing machine produced by Jinan Times (Jinan, China), as shown in Figure 2. The maximum test force is 300 kN and the resolution of beam displacement is 0.001 mm. The test equipment is widely used in the mechanical property test of metal and non-metal such as pulling, pressing and bending.

### 2.3. AE Monitoring Setup

The acoustic emission instrument used in the experiments was a PCI-2 acoustic emission system manufactured by Physical Acoustics Corporation (PAC), (Princeton Junction, NJ, USA), in which the acoustic emission system mainly consisted of acoustic emission sensors, preamplifiers, and acoustic emission acquisition boards. The sensors used in the acoustic emission signal acquisition test for tensile damage include two R15a-type acoustic emission sensors. Two 2/4/6 type three-position preamplifiers are connected to the acoustic emission sensors, which amplify the acoustic emission signals’ output by the sensors and then supply them to the computer through the cable for analysis and processing. The role of the PCI data acquisition card is to convert the analogue electrical signals into digital signals through sampling and processing and then transmit them to the computer processing system, which then performs the calculation and processing process. An appropriate amount of petroleum jelly is used evenly as coupling agent on the contact surface between the acoustic emission sensor and the specimen, and the sensor is fixed to the specimen with adhesive tape. The acoustic emission instrument is tuned on and the acoustic emission testing application software AEwin entered (PCI2-E5.90, Princeton Junction, NJ, USA) in which the settings of the acoustic emission test parameters are completed. In order to reduce the appearance of noise signals caused by the friction generated between the clamp chuck and the specimen, the specimen was therefore preloaded before the test started. After the field measurement of the ambient noise, the threshold value of the test settings 1 and 2 channels was 47 dB. Amounts of 300 μs, 600 μs, and 1000 μs were chosen as the peak definition time (PDT), hit definition time (HDT), and hit lockout time (HLT), respectively. The sensitivity of the acoustic emission detection equipment will greatly affect the results of damage detection, in order to ensure the reliability of the test results, before the start of the test by conducting a lead break test to calibrate the sensitivity of the acoustic emission detection equipment. When the maximum difference in the amplitude of the acoustic emission signal detected by the acoustic emission sensor at a specific point is less than 3 dB, the sensor sensitivity meets the requirements. Firstly, an acoustic emission sensor was arranged at one end point of the specimen, and 10 lead breakage tests were performed at 50 mm, 100 mm and 150 mm from the sensor by using a type 2B automatic pencil with a lead breakage length of 3 mm, and the inclination angle between the pencil and the surface of the material at the time of lead breakage was 30°. The results of the signal amplitude of the 10 tests done at different positions are shown in Figure 3. From Figure 3, it can be seen that the maximum amplitude difference at the same lead break position is less than 3 dB, whether at 50 mm or 100 mm or 150 mm from the sensor, which indicates that the sensitivity of the acoustic emission sensor meets the test accuracy requirements. Figure 4 illustrates the acquisition of the signal. An appropriate amount of petroleum jelly is used evenly as coupling agent on the contact surface between the acoustic emission sensors and the specimen, and the sensors are fixed to the specimen with adhesive tape to collect the signals.

## 3. Analysis and Discussion

### 3.1. Analysis of Acoustic Emission Signal Parameters

When a metal material (or component) is subjected to a load, the result of the damage process is reflected in the deterioration of the relevant mechanical properties of the metal material, and is accompanied by the release of strain energy, part of which is released in the form of stress waves, thus producing acoustic emission. Therefore, the acoustic emission phenomenon occurs during the stages of damage change such as yielding and fracture, and the strain energy released is different due to the difference of damage mechanism in different stages, and the characteristics of acoustic emission signal also appear. The method of acoustic emission parameter experience analysis is to analyze the trend of acoustic emission signal parameters with damage evolution time in order to obtain information about the activity and trend of acoustic emission damage source in each damage stage. By analyzing the characteristics of the acoustic emission signal generated by the damage process, it is possible to visualize the trend of the test process signal, the stage of the damage, and certain distinctive damage characteristic signals.

The amplitude is the largest amplitude value in the acoustic emission signal waveform, and is often used to identify the source type, determine the intensity of the source, and measure the attenuation. Figure 5 is a graph of amplitude versus load over time, and it is obvious that the amplitude of the signal varies greatly in different damage stages. In the elastic stage, the amplitude of the acoustic emission signal is low, almost always below 50 dB, and there is no tendency to decrease or increase. Additionally, from the elastic stage to the transition of the plastic yielding stage, the amplitude begins to show a significant growth trend. Until the whole yielding stage, the amplitude reaches a very high level, and even a signal close to 95 dB appears. In the strengthening stage, the amplitude is at a high level in the early stage, and the amplitude starts to decrease as the strengthening stage progresses, but the overall amplitude is higher than that of the elastic stage. In the necking stage before fracture, the amplitude is almost zero until the moment of fracture, when the highest amplitude of nearly 100 dB occurs during the entire damage process. In summary, the trend of amplitude changes throughout the damage process from a very low amplitude stage (corresponding to the elastic stage) to a large increase in amplitude (corresponding to the yielding stage), then through a gradually decreasing stage (corresponding to the strengthening stage) to almost no amplitude (the necking stage before fracture), and finally to the highest amplitude point in the process (corresponding to the fracture moment).

Ringing count is the number of times a ringing pulse crosses the threshold voltage. The ringing count is related to the energy of the event generating the acoustic emission, and it can roughly represent the intensity and frequency of the signal, which is applicable to both sudden and continuous signals. Thus, it is very useful in the evaluation of acoustic emission, and the ringing count is very sensitive to the deformation damage of the material. As a result, the damage process was analyzed by the change of acoustic emission ringing counts with time. Figure 6 shows the history of ringing counts in the damage process. In the elastic stage, the overall change in the ring count is not significant, while the ring count shows an increasing trend after entering the plastic yielding stage. When entering the plastic yielding process, a large number of dislocations start to move due to the stress concentration, accompanied by the release of energy, which leads to the acoustic emission signal with a high ringing count value at this time. In the strengthening stage, the internal dislocation density increases, dislocations interact with each other, and dislocations are blocked, entangled and fixed, thus causing obstacles to the dislocation movement. In the pre-strengthening phase, all slip bands are stretched in the tension direction, so there are no acoustic waves corresponding to the deformation of the slip bands during this phase. As the strengthening phase progresses, the number of movable dislocations decreases, making it difficult to continue the plastic deformation, and their activity decreases significantly compared to the pre-strengthening phase, thus the tendency of the acoustic emission parameters to decrease during this phase is related to the decrease in dislocation activity. The transient nature of the signal leads to a short duration, which results in small values of the ringing counts. The specimen is necked before fracture, and when the cross-sectional area of the specimen decreases after necking, the load capacity decreases, and the force curve shows a smooth downward trend. At the moment of fracture, the internal dislocations of the material move violently and are accompanied by great vibrations due to the fracture, but the plastic deformation of the specimen is greatly limited by the fracture time. However, because the fracture time is very short, the change in parameters is not significant.

Since Q345 steel is a polycrystalline metal material, the deformation phenomenon occurs mainly due to the gradual movement of dislocations along the slip surface. When the dislocations move at a high velocity, the local stress field near the dislocations generates acoustic emission signals. When entering the plastic yielding process, a large number of dislocations start to move due to the stress concentration, which is accompanied by the release of energy, resulting in the acoustic emission signal with high amplitude and high ringing count value at this time. In addition, when the dislocations start to move in order to achieve plastic deformation, they accelerate and proliferate, causing the fragmentation of grains and resulting in an increase in dislocation density. When the damage is close to the yield point, the movable dislocations reach their highest point, resulting in an acoustic emission source event with high activity characteristics.

### 3.2. Spectrum Analysis

In order to obtain the characteristics of the signal in the frequency domain during this acoustic emission tensile test phase, the spectral characteristics of the acoustic emission signal at the moment of yielding and fracture during tensile damage were analyzed using the fast Fourier transform.

The typical signals with more obvious changes such as the yielding stage and fracture moment as well as impact noise were extracted, respectively, in which the waveform diagram of the yielding damage stage is shown in Figure 7, in which the signal behaves as a mixed type signal. Figure 8 shows the spectrum of the signal in the yielding stage, and the signal extracted in the plastic yielding stage appears with two obvious peak frequencies of 170 kHz and 250 kHz.

Figure 9 and Figure 10 show the signal waveform and spectrum at the moment of fracture, respectively, from which it can be found that the peak frequency of the signal at the fracture stage is 138 kHz.

Figure 11 shows the extracted noise waveform plot. Figure 12 shows the spectrum of the noise signal, with the peak frequency of the impact noise signal around 31 kHz.

From the time domain waveforms of the typical signals of the above three different damage stages, it can be found that the voltage amplitude values of the noise signals are much lower than those of the other two types of signals. In the acoustic emission signals at the yielding and fracture moments of tensile damage, the voltage amplitude values of the signal at the yielding stage are much smaller than the values at the fracture moment. The frequency of the noise signal was found to be below 100 kHz, while two peaks were evident at 170 kHz and 250 kHz during the yielding phase. The peak frequency at the time of fracture is 138 kHz and its frequency band is wider.

### 3.3. Acoustic Emission Normalized Cumulative Parameter Analysis

Since the definition and magnitude of different acoustic emission parameters are different, there are some differences in the analysis by the variation of different cumulative parameters of acoustic emission signals. In this section, in order to compare the accumulated parameters of different types of acoustic emission to obtain the characteristics of the accumulated parameters of each acoustic emission in different damage stages with time, the normalized data processing is performed for the accumulated parameters of each acoustic emission signal. After the normalized data processing, the respective normalized cumulative parameter curves under three different loading rates and the changes of cumulative parameters between different loading rates were compared and analyzed, so that the occupancy of each characteristic parameter in different damage stages could be visually observed.

The purpose of parameter normalization is to analyze the proportion of each component to the overall volume. Thus, the following formula is used for data normalization.
(1)$$Y=\frac{X-X_{min}}{X_{max}-X_{min}}$$
where: Χ  and Y are the data of acoustic emission accumulation parameters before and after the normalization process, while $X_{max}$ and $X_{min}$ are the maximum and minimum values of the acoustic emission accumulation parameters X, respectively.

The most important feature of this normalization method is that it does not change the proportion of each cumulative acoustic emission parameter in the cumulative total, thus preserving the trend of each cumulative characteristic parameter curve. Moreover, the sensitivity of different cumulative parameters to different types of damage can be reflected by the increase in each cumulative acoustic emission parameter at each damage stage, so the normalized cumulative acoustic emission parameters are more suitable for analyzing the acoustic emission activity at different stages of the whole damage process. In the comparative analysis of the changes of each cumulative characteristic parameter under the same loading rate, the normalized cumulative acoustic emission parameters of the tensile specimen with a loading rate of 1 mm/min are analyzed as an example, as shown in Figure 13a. The normalized parameters include: the normalized cumulative amplitude parameter, normalized cumulative count parameter, normalized cumulative energy parameter, normalized cumulative duration parameter and normalized cumulative hit parameter.

From Figure 13a we can find that, in terms of the trend of each cumulative parameter, each cumulative parameter shows a certain linear growth trend in the elastic phase. Since Q345 steel is a plastic material, it can be found from Figure 13a that there is a clear yielding stage, and each normalized cumulative parameter in the yielding stage has a jumping growth until the beginning of the strengthening stage, which gradually changes to a slow growth phenomenon. However, it is worth noting that the cumulative amplitude and cumulative hit parameters show a significant growth trend at the beginning of the strengthening phase until the necking phase when the growth rate decreases. After the necking stage, the normalized cumulative parameters almost stop growing until the moment of fracture, when another jump in growth occurs. The jump–growth curves at the end of the yielding stage and at the moment of fracture indicate that there is a large amount of accumulation of acoustic emission parameters occurring at this moment, and these two points are found to be almost identical to the transition points delineated by mechanical curves. The normalized cumulative amplitude and the cumulative hits change curves are almost identical throughout the process, and they have a high similarity. From Figure 13b,c, we can see that the hits numbers are mostly distributed in the plastic yielding phase, followed by a larger distribution in the elastic phase, and a small number in the remaining phases. The normalized cumulative hits and hits per second have a significant increase around 300 s, which is due to the deformation that occurs in the specimen as the volume of the material increases. In contrast, the number of hits tends to decrease significantly as the yielding phase enters the strengthening phase, and the number of hits is small until the moment of fracture.

From Figure 13a we can find that in terms of the proportion of each cumulative parameter component, the normalized cumulative energy parameter differ from the other cumulative parameters in the two phases where a clear trend change occurs (yielding phase and fracture moment). The proportion of cumulative energy in the yielding phase is much smaller than that in the fracture moment, which indicates that the cumulative energy parameter is more sensitive to the occurrence of fracture damage than other cumulative parameters. The proportion of other cumulative parameters in the yielding phase is much higher than in the fracture moment, especially the cumulative count and duration parameters change more significantly, which indicates that the cumulative count and duration parameters are more sensitive to the occurrence of damage in the yielding phase. The reason for this is that a certain degree of plastic deformation occurs, resulting in a large number of movable dislocations, which leads to a large number of acoustic emission signals at the end. The increase in cumulative duration during the yielding phase was the highest, reaching nearly 80%, and the increase in cumulative counts reached nearly 55%; however, the increase in cumulative amplitude and cumulative impact were both at low levels, near 30%, compared to the change in the percentage of cumulative duration and cumulative counts. However, at the end of the yielding phase and the beginning of the strengthening phase, the growth of cumulative hit and cumulative amplitude increased rapidly, close to 40%, while the proportion of cumulative count and cumulative duration parameters was less, which indicated that the cumulative amplitude and cumulative hits parameters were more distinctive in the strengthening phase and sensitive to the damage which occurred in the strengthening phase.

The normalized cumulative energy parameter curve increases the most at the moment of fracture. The reason for this is that the increase in local deformation during the necking period leads to a very high level of material slip accumulation within the necking area, where the dislocation plugging becomes so severe that a jump in the accumulated energy of the acoustic emission signal occurs at the time of fracture. In the yield damage stage, plastic deformation gradually occurs, which is microscopically manifested in the dislocation and deformation of the grains inside the specimen, resulting in a significant jump in the normalized cumulative parameters at the completion of the yield damage, also due to the presence of a very large number of movable dislocations. The analysis of the normalized cumulative parameters led to a more in-depth understanding of the sensitivity of different parameters at different damage stages, and it was found that the cumulative duration and cumulative count parameters were sensitive to the onset of yield damage, while the cumulative amplitude and cumulative hit parameters were sensitive to the strengthening damage stage, and the cumulative energy parameters were sensitive to the final fracture moment.

### 3.4. Characteristics of Mechanical and Acoustic Emission Signal Changes under Different Loading Rates

Based on the results of the above analysis of the variation differences between the individual normalized cumulative parameters during the damage process, the differences in the variation of the normalized cumulative parameters are compared here by comparing the tensile rate conditions of 1 mm/min, 1.5 mm/min and 2 mm/min, respectively, to obtain the characteristics of the different cumulative parameters.

#### 3.4.1. Changes in Mechanical Properties for Different Loading Rates

Three sets of tensile tests of Q345 specimens at different rates were carried out, and the load displacement curves were obtained at tensile rates of 1 mm/min, 1.5 mm/min and 2 mm/min, respectively, as shown in Figure 14. From the comparison results, it can be seen that the three loading rates in the test do not have much influence on the tensile curves.

#### 3.4.2. Variation of Tensile Force and Number of Hits at Different Loading Rates

Figure 15 shows the distribution of the number of acoustic emission hits during the whole damage process, and the distribution of the number of hits has obvious characteristics.

The hits are mostly distributed in the plastic yielding stage, followed by a larger distribution in the elastic stage, while the number in the remaining stages is small. For the elastic stage acoustic emission signal parameters vary with time, the reason is that the hits generated in the elastic stage are mostly due to the noise caused by the friction between the machine and the specimen. The high number of hits generated during the yielding plastic deformation of the specimen is due to the deformation that occurs in the specimen as the volume of the material increases. In the yielding stage, the number of hits tends to decrease when entering the strengthening stage, and the number of hits is very small until the fracture moment, which is very similar to the characteristics of the acoustic emission parameters over time. At the moment of fracture, the number of hits is not high due to the large amount of strain energy released in a split second, while the signal at this moment is of the burst type.

#### 3.4.3. The Normalized Cumulative Parameters of Acoustic Emission at Different Loading Rates

The cumulative hit, cumulative amplitude, cumulative energy, cumulative count and cumulative duration parameters of the acoustic emission at three different rates were normalized, and the normalized cumulative parameters of the acoustic emission at rates of 1 mm/min, 1.5 mm/min and 2 mm/min are plotted in Figure 16, Figure 17 and Figure 18, respectively.

From the analysis of the trend of the normalized cumulative parameters, the trend of the normalized cumulative parameters at the three different tensile rates was more or less the same at different damage stages. In particular, the normalized cumulative parameters showed a sudden increase in the yielding stage and the fracture moment, and the cumulative energy parameters all showed a large increase at the fracture moment. Moreover, the curves of cumulative amplitude and cumulative impact at each damage stage almost coincide regardless of the rate change. However, the proportion of the cumulative parameters and the amount of increase at different stages varied with the change in rate. The most obvious of these is the change in cumulative duration, where the incremental decrease in cumulative duration with increasing tensile rate occurs abruptly at the yielding stage, decreasing from nearly 80% incremental at the tensile rate at 1 mm/min to nearly 50% at the 1.5 mm/min rate, to 45% incremental at the final 2 mm/min rate. However, the percentage of increments in the cumulative duration after the beginning of the strengthening phase shows a tendency to increase with the increase in the stretching rate and tends to be chaotic during the necking phase. Since the end of the plastic yielding stage coincides with the transition point of the specimen to the strengthening stage, the cumulative duration at relatively low loading rates can visualize the transition from incomplete plastic deformation to full plastic deformation in the tensile damage process. However, the proportional component of the cumulative energy parameter at each stage is almost independent of the tensile rate, indicating the potential of the cumulative energy parameter as a characterization of damage. Similarly, the change in the cumulative count parameter and the proportional component of the cumulative energy parameter at each stage did not change significantly regardless of the change in tensile rate.

In combination with the analysis of the variation of each cumulative parameter for a specific tensile rate in the previous section, it was found that the cumulative count parameter is less affected by the tensile rate and has the potential to characterize the occurrence of damage at the yielding stage. Similarly, the cumulative energy parameter hardly changes with tensile rate and is very sensitive to the moment of fracture, making it useful for characterizing the moment of fracture. The cumulative impact and cumulative amplitude, on the other hand, are more sensitive to the occurrence of damage during the strengthening phase.

#### 3.4.4. Inter-Correlation Analysis

From the results of the above analysis, it is clear that acoustic emission signal characteristics parameters can be used to characterize the material damage. However, there are multiple acoustic emission parameters that can be used to characterize the damage, and each parameter characterizes the damage process differently. In order to analyze the connection between the individual acoustic emission cumulative parameters, the correlation method was used to understand the degree of correlation between the trends of each cumulative parameter at each damage stage.

The interrelationship number can be used to express the degree of correlation between the changes in the parameters of the two comparison variables and is defined by the formula shown below.
(2)$$\;\rho_{xy}=\frac{Cov\left(x,y\right)}{\sqrt{D\left(x\right)}\sqrt{D\left(y\right)}}=\frac{E\left[\left(x-u_{x}\right)\left(y-u_{y}\right)\right]}{\sqrt{E\left[{\left(x-u_{x}\right)}^{2}\sqrt{E\left[{\left(x-u_{y}\right)}^{2}\right]}\right]}}$$

$Cov\left(x,y\right)$ is the covariance of the random variables, $D\left(x\right)$ and $D\left(y\right)$ are the variances of the random variables x and y, and $u_{x}$ and $u_{y}$ are the means of the random variables  x and y. The value of $\;\rho_{xy}$ is between −1 and 1. A smaller value indicates less similarity in the variation of the parameters of the two compared variables.

In order to obtain the correlation between different acoustic emission accumulation parameters during the whole tensile damage process, the changes of different acoustic emission accumulation parameters during the damage process were investigated by means of correlation analysis. In order to compare with the analysis of each acoustic emission accumulation parameter in the previous subsection, the cumulative duration, cumulative energy, cumulative impact, cumulative amplitude, and cumulative count were analyzed by mutual correlation analysis, and finally, the mutual relationship values between each acoustic emission accumulation parameter were calculated. The results are shown in Table 2, where the obtained correlation values reflect the degree of similarity of the damage trends used to characterize the curves of each acoustic emission accumulation parameter over time.

From the results in Table 2, it can be found that the correlation coefficients of the curves of each cumulative characteristic parameter in the damage process of Q345 steel are very high. The correlation coefficients of all the cumulative parameters exceed 0.98, and most of them reach 0.99 or more. This indicates that the trends of the curves of the cumulative parameters in the tensile damage process of Q345 steel are close to each other, and, to a certain extent, the changes of the cumulative acoustic emission parameters can be used to characterize the changes of the tensile damage process. The curves of the cumulative acoustic emission parameters and the normalized cumulative parameters are more suitable for the analysis of Q345 steel tensile deformation damage process. The correlation analysis of the variation curves of the acoustic emission accumulation parameters again shows that the good plasticity of Q345 steel makes the variation of the acoustic emission accumulation parameters similar to each other, and it also shows that the analysis and evaluation of the tensile damage process by the acoustic emission accumulation parameters is more reliable.

## 4. Conclusions

In this paper, the characteristics of acoustic emission signals generated during tensile damage of wind turbine tower material Q345 steel were analyzed. The following conclusions were drawn.

The changes of acoustic emission signals and parameters during the damage process and the signal frequency characteristics at the yielding and fracture moments were investigated. It was found that from the transition point between the elastic stage and the yielding stage, the number of acoustic emission signals increased abruptly and the signal amplitude and counting parameters all showed a tendency to increase significantly, and the parameter values of the signals remained high until the end of the yielding stage. After the beginning of the strengthening phase, the signal parameters showed a decreasing trend until the necking phase when almost no signal was generated. Very few signals with very high parameter values appear at the moment of fracture. The signal in the yielding phase has two distinct peak frequencies in the frequency domain, 170 kHz and 250 kHz, while the peak frequency of the signal in the fracture moment is 138 kHz and the peak frequency of the noise signal is 31 kHz.The variation of the normalized acoustic emission accumulation parameters during the damage process was investigated, and it was found that the normalized accumulation parameters underwent a very obvious sudden increase at the transition point from the end of the elastic phase to the yielding phase and at the moment of fracture, but with a large difference in the increase. At the transition point from the elastic stage to the yielding stage, the proportional increase in the normalized cumulative duration was the largest, reaching 80%. In contrast, the increment in normalized cumulative energy is the lowest, at 8%. Therefore, the normalized cumulative duration parameter is suitable for characterizing the yield damage occurrence. At the moment of fracture, the normalized cumulative energy parameter increased by 90%, while the other normalized cumulative parameters changed by less than 10%, indicating that the normalized cumulative energy parameter is very sensitive to the occurrence of fracture.The effects of three different loading rates on the normalized cumulative parameter and the number of hits were investigated, and the results showed that the normalized cumulative energy parameter was least affected by the loading rate, and the energy parameter was a sensitive factor for the normalized expression. The proportional increment of the normalized cumulative duration parameter at the transition between the elastic and yielding stages decreases with increasing rate. As the loading rate increases, the smoothness of the curves of each normalized cumulative parameter in the strengthening phase becomes less and less smooth. The growth rate of the normalized cumulative energy change starts to show a slight increasing trend with the increase in the test loading rate during the period from the beginning of the yielding phase until the occurrence of fracture. During the yielding plastic deformation of the specimen, a higher number of hits are generated due to the deformation of the specimen as the volume of the material increases. In the yielding phase, after entering the strengthening phase, the number of hits tends to decrease until the moment of fracture, when the number of hits is very low, which is very similar to the characteristics of the acoustic emission parameters with time. At the moment of fracture, the number of hits is not high due to the instantaneous release of a large amount of strain energy.

In this paper, we studied the changes of acoustic emission signals and parameters during damage and the signal frequency characteristics at the yielding stage and fracture moment in Q345 steel in tensile tests. We analyzed the changes of the acoustic emission cumulative parameters during the damage process by normalizing the parameters, and the effects on the normalized cumulative parameters were studied and discussed at three different loading rates. The study of whether the normalized cumulative parameters change significantly under the influence of different materials and external conditions needs to be further investigated. Further depth is needed in the quantitative study of material damage by acoustic emission parameters to establish a deeper correlation between acoustic emission parameters and the degree of material damage. In future work, we could analyze the b-value to evaluate the degree of damage to the material.

## Figures and Tables

**Figure 1 materials-14-02120-f001:**
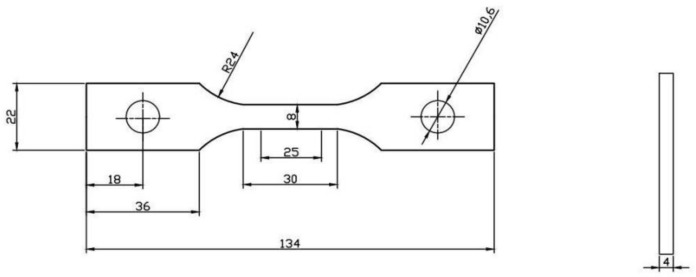
Q345 steel specimen size.

**Figure 2 materials-14-02120-f002:**
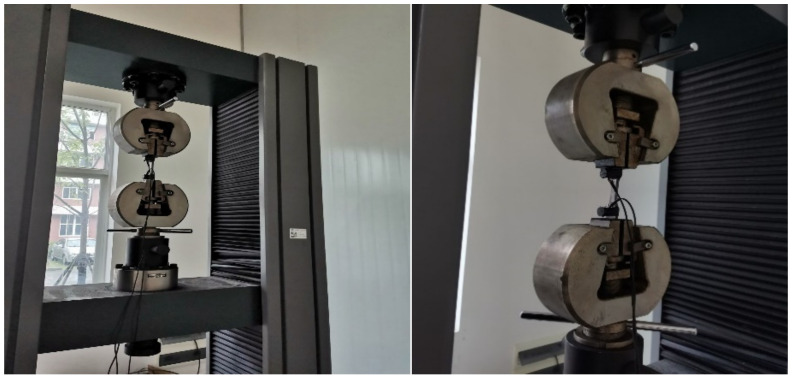
Tensile testing equipment.

**Figure 3 materials-14-02120-f003:**
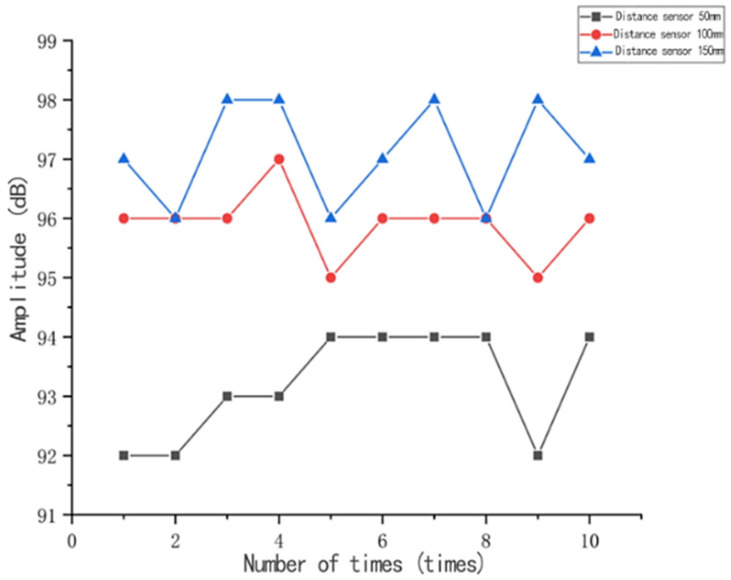
Graph of the results of the lead break test.

**Figure 4 materials-14-02120-f004:**
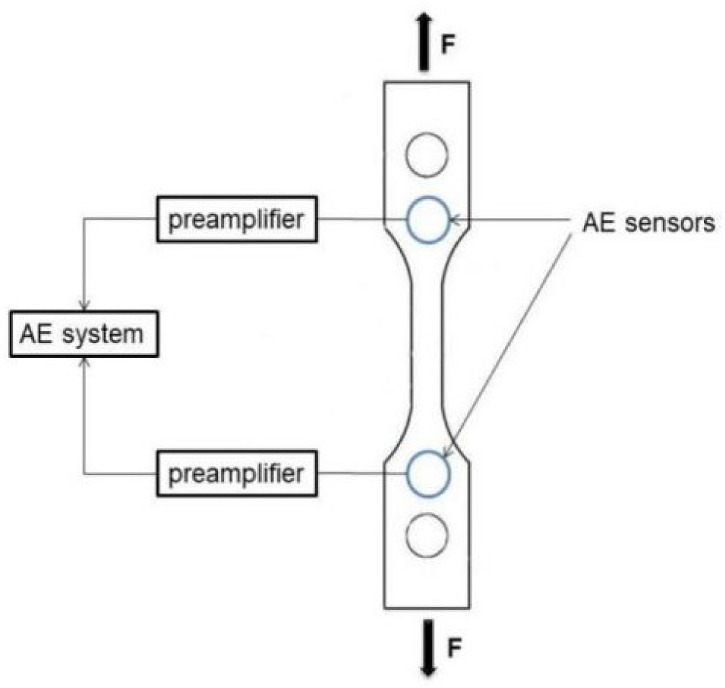
Schematic diagram of signal acquisition.

**Figure 5 materials-14-02120-f005:**
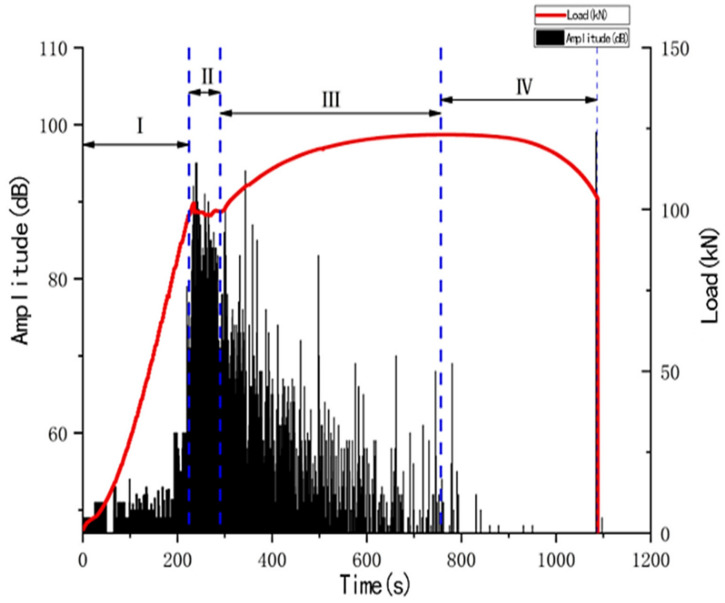
Signal amplitude parameters over time.

**Figure 6 materials-14-02120-f006:**
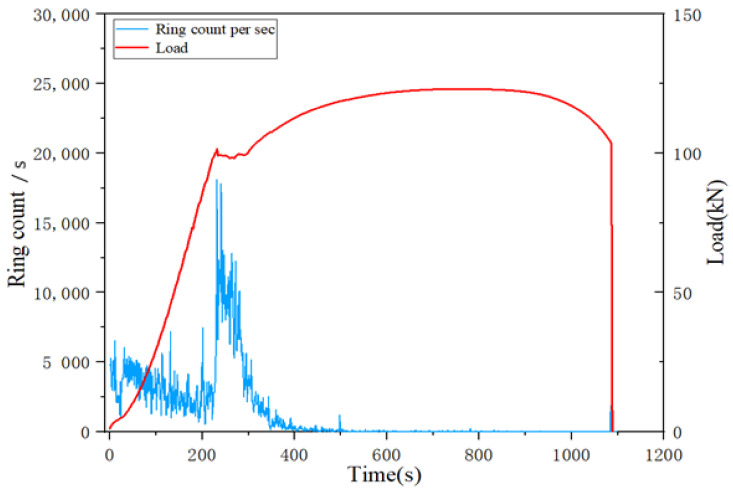
Signal ring count changes during the damage process.

**Figure 7 materials-14-02120-f007:**
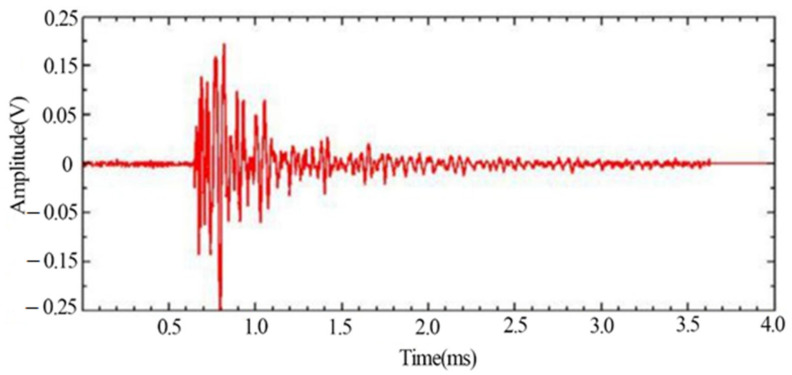
Typical acoustic emission signal waveform in the yielding stage.

**Figure 8 materials-14-02120-f008:**
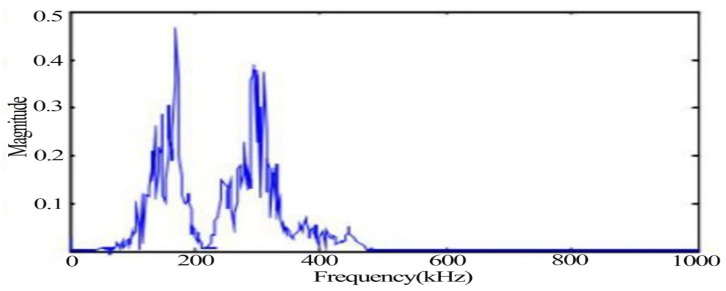
Typical acoustic emission signal spectrum in the yielding stage.

**Figure 9 materials-14-02120-f009:**
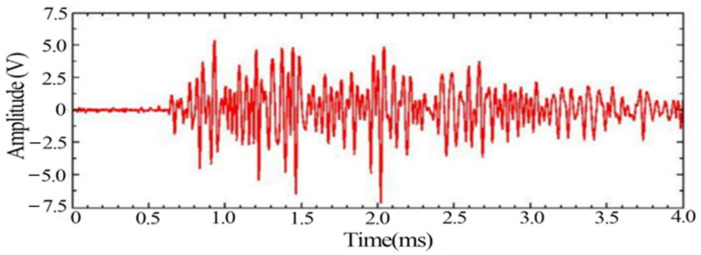
Signal waveform at the moment of fracture.

**Figure 10 materials-14-02120-f010:**
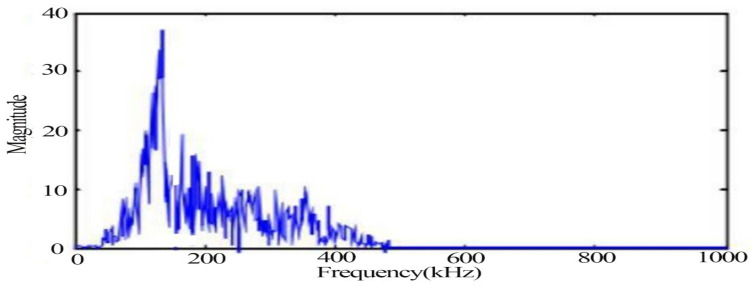
Signal spectrum at the moment of fracture.

**Figure 11 materials-14-02120-f011:**
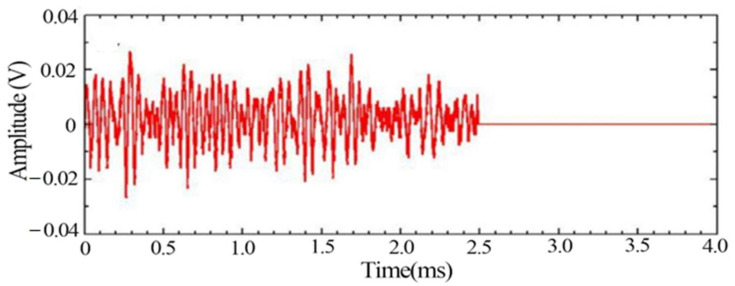
Waveform of noise signal.

**Figure 12 materials-14-02120-f012:**
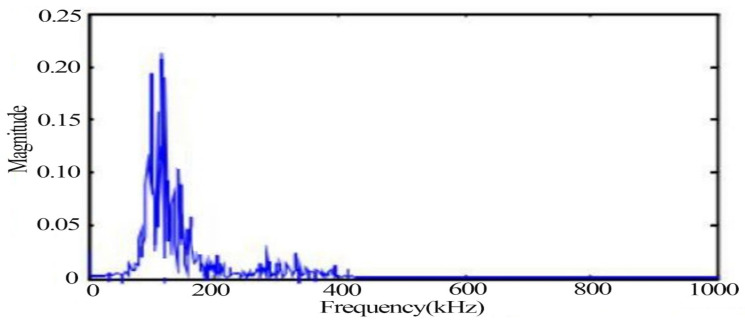
Spectrum of noise signal.

**Figure 13 materials-14-02120-f013:**
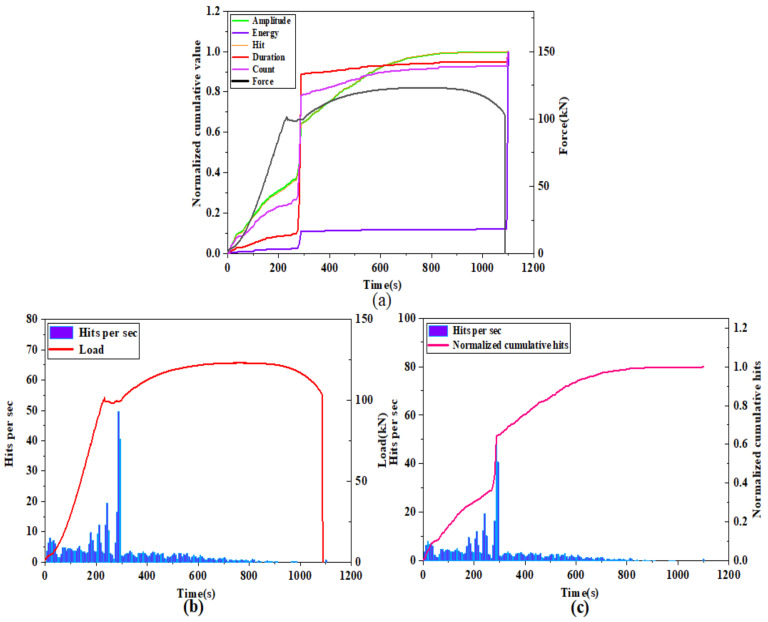
(**a**): change in normalized cumulative parameters during tensile damage; (**b**): change in number of hits per second and tensile force during tensile damage; (**c**): change in normalized cumulative number of hits and number of hits per second during tensile damage.

**Figure 14 materials-14-02120-f014:**
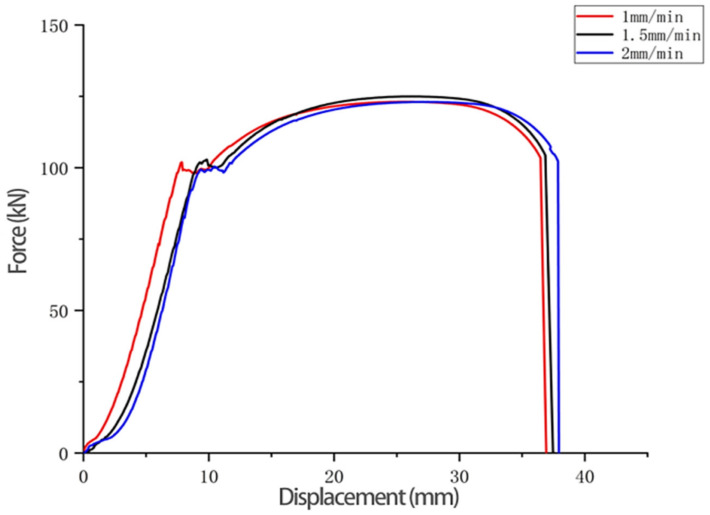
Tensile force and displacement curves of Q345 specimens at different rates.

**Figure 15 materials-14-02120-f015:**
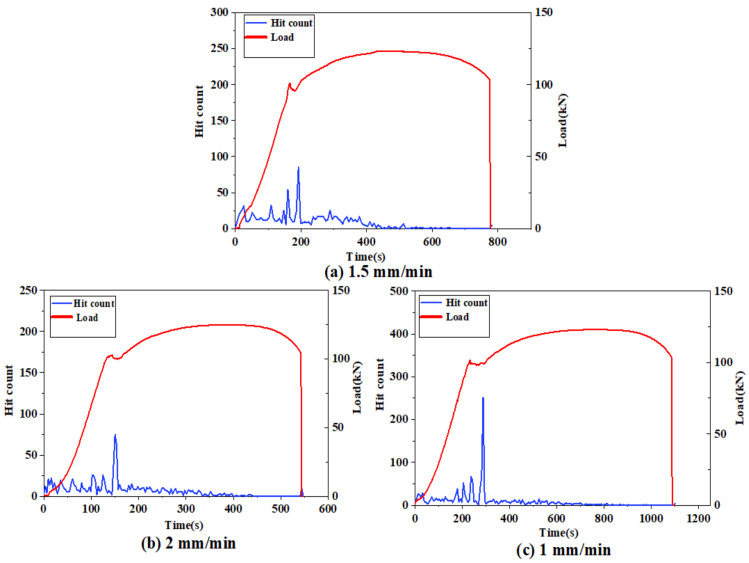
(**a**): variation of hit counts with stretching force for a rate of 1.5 mm/min; (**b**): variation of hit counts with stretching force for a rate of 2 mm/min; (**c**): variation of hit counts with stretching force for a rate of 1 mm/min.

**Figure 16 materials-14-02120-f016:**
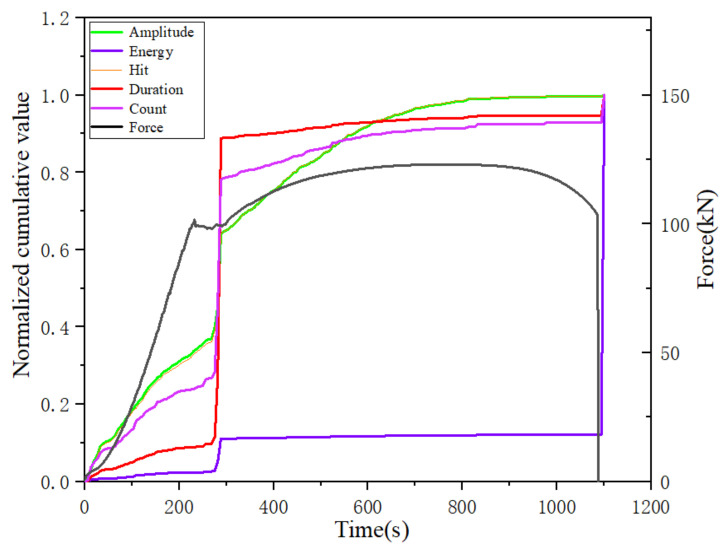
Variation of each normalized cumulative parameter at a loading rate of 1 mm/min.

**Figure 17 materials-14-02120-f017:**
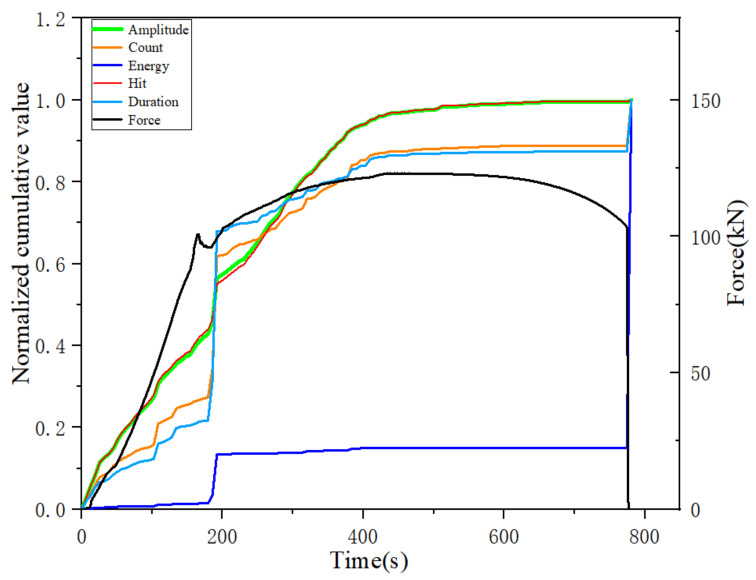
Variation of each normalized cumulative parameter at the loading rate of 1.5 mm/min.

**Figure 18 materials-14-02120-f018:**
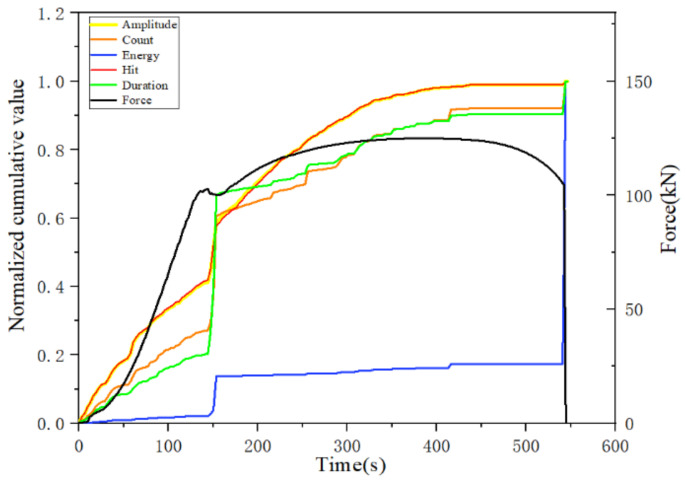
Variation of normalized cumulative parameters at the loading rate of 2 mm/min.

**Table 1 materials-14-02120-t001:** Chemical compositions (wt.%) of Q345 stainless steel.

Composition	C	Si	Mn	S	P
wt.%	0.11–0.20	0.20–0.54	1.20–1.60	<0.045	<0.045

**Table 2 materials-14-02120-t002:** Interrelationship values between different acoustic emission accumulation parameters.

Parameters	Amplitude	Count	Energy	Duration	Hit
Amplitude	1	0.9926	0.9838	0.9943	0.9843
Count	0.9926	1	0.9951	0.9993	0.9921
Energy	0.9838	0.9951	1	0.9963	0.9854
Duration	0.9943	0.9993	0.9963	1	0.9956
Hit	0.9843	0.9921	0.9854	0.9956	1

## Data Availability

Data is contained within the article.

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
