# Peer review of "Tensile Damage Study of Wind Turbine Tower Material Q345 Based on an Acoustic Emission Method"

_materials, 2021, doi:10.3390/ma14092120_

Round 1
Reviewer 1 Report
Manuscript ID: materials-1168197
Type of manuscript: Article
Title: Tensile damage study of wind turbine tower material Q345 based on acoustic emission method
Authors: Xiao Tang, Lida Liao *, Bin Huang, Cong Li
In this work, a simple tensile curve was studied. Q345 steel, used in the manufacture of wind turbine towers, was examined in this study. The authors should be clearer in their contributions, what is the originality of this work. This paper's aims are unclear. They purport to show a new method of AE analysis, but the analysis they use is at least 40 yrs old and they apply wrongly without establishing the black-box idea to real damage connection. There is little I can commend and this should be rejected.
Please consider the below.
The authors should refer to the references below which conducted AE measurement on steel.
- P.M. Fernando, D.P. Miguel, R.M. Luis, Performance analysis of acoustic emission hit detection methods using time features, IEEE Access 7 (2019) 71119–71130.
- F.F. Barsoum, et al., Acoustic emission monitoring and fatigue life prediction in axially loaded notched steel specimens, J. Acoustic Emission 27 (2009) 40–63.
- C. Ennaceur, et al., Monitoring crack growth in pressure vessel steels by the acoustic emission technique and the method of potential difference, Int. J. Press. Vessels Pip. 83 (3) (2006) 197–204.
Reviewer 2 Report
The paper is an elaborated piece of research on the application of AE methods to study damage of Q 345 steel used in wind turbine towers. What is needed according to the present review is the correlation between the signals of the AE method applied and the loading conditions of the structure to be clearly and in details explained.
Reviewer 3 Report
The paper discusses the study of the tensile damage of wind turbine tower material Q345 in the light mainly of the Acoustic Emissions technique and normalized accumulation parameters.
This work is quite interesting and aims to investigate the detection of criticality of the damages occurring in Q345 steel materials for wind turbine tower construction.
I would like some clarifications to be given and I enclose some suggestions that the authors should take into account in a revised submission.
In Figure 5 the graph of amplitude versus load over time is presented. The authors could extend or focus on the b-value analysis, a popular method for evaluating the degree of damage of materials, [see: Shiotani T. Application of AE improved b-value to quantitative evaluation of fracture process in concrete materials. J Acoustic Emission, 2001; 19: 118-133 and Loukidis A., et al., Comparative Ib-value and F-function analysis of Acoustic Emissions from elementary and structural tests with marble specimens, Material Design and Processing Communication, 2020; e176].
In Figure 6 the authors present the acoustic activity using the Ring Down Count (RDC) parameter. Do the values shown in the figure refer to the RDC rate (RDC per sec)? This needs to be made clear in the text and in the figure. The authors could also use the parameter of AE hits rate [hits per sec].
A better presentation in Figures 13 and 15-17 is required. I suggest three Figures that will correspond to loading rate 1.0, 1.5 and 2 mm/min in which the tensile force is presented on the secondary axis.
It would be interesting to have a comparison of the acoustic activity with the three loading rates, since in Figures 15-17 the sizes are shown in normalized scale. I would suggest that the authors present three diagrams [1.0, 1.5 and 2 mm/min] versus time, with the primary axis corresponding to the hits per sec and the secondary axis to the tensile force. It would also be interesting to render the AE activity with more clarity when the tensile force presents the first immersion to render. The authors could adopt a new way of presenting acoustic activity (see: "An Alternative Approach to Representing the Data Provided by the Acoustic Emission Technique, Rock Mechanics and Rock Engineering (2018) 51: 2433–2438"). If the authors do not wish to study AE activity according to the above suggestion in this work, they could attempt to present a continuous representation of RDCs per sec, taking the sliding mean value of RDCs of a group (e.g., N = about 10) consecutive hits and corresponding to this value as time, the average time of the events of N successive AEs. In this way, a continuous time capture of RDC per sec will be achieved and a comparison for the three loading rates will be provided.
Finally, in the Introduction, the sentence: “Acoustic emission detection technology have been widely used in damage detection including for metallic materials [13, 14], composite materials [15, 16], and concrete materials [17, 18]”, should be revised. References 17, 18 refer to rock materials not, concrete materials. It is recommended to add the references: SK Kourkoulis, et al., Notched marble plates under tension: Detecting pre-failure indicators and predicting entrance to the “critical stage”, Fatigue & Fracture of Engineering Materials & Structures, 41, 776-786 (2018) and SK Kourkoulis, et al., Notched marble plates under direct tension: Mechanical response and fracture, Construction & Building Materials, 167 426–439 (2018), where the AE method is applied, in marble specimens under tensile force with specimen geometry similar to that of the present work.
Reviewer 4 Report
The paper investigates the damage evolution process in the tensile loading of Q345 steel specimens under different loading rates. The authors used AE in combination with time-domain and frequency-domain analysis to distinguish different damage evolution stages and also to identify the source of different AE signals. The topic is interesting and the authors performed valuable experimental works. However, the paper is not organized very well and the novelty of the work is hidden. Because, from the AE analysis point of view, there is no new analysis method and it seems that just the application field is new. There are some comments that should be considered during the resubmission:
• The introduction is not coherent with the main topic of the work and the novelty has not been highlighted. Please modify it.
• There are some typos, please revise the language of the manuscript carefully.
• Page 2: “Acoustic emission detection technology have been widely used in damage detection including for metallic materials [13, 14], composite materials [15, 16]…” There is a recently-published review paper on the damage characterization of composite materials using AE that covered the main challenges of the topic. It is good if you refer the readers to it as well.
Damage characterization of laminated composites using acoustic emission: A review. Composites Part B: Engineering, Volume 195, 2020, 108039, https://doi.org/10.1016/j.compositesb.2020.108039
• Figure 1: what are the two circles with a diameter of 10.6 mm at both sides of the tensile specimen?
• Figure 2: please insert an image of the test setup that shows tensile machine, specimen, AE sensors, etc.
• Page 4: Why did you select PDT and HDT parameters too high?
• Page 4: What AE threshold value did you use for recording AE events?
Round 2
Reviewer 1 Report
Accept in present form
Author Response
Dear editor Victor Ibanescu and reviewers: On behalf of my co-author, thank you very much for providing us with the opportunity to revise the manuscript, and thank the editors and reviewers for their positive and constructive contributions to our manuscript entitled "Tensile Damage Research on Wind Turbine Towers" Comments and suggestions. Material Q345" based on acoustic emission method. (ID: materials-1168197). We have carefully studied the reviewer’s comments, revised them, and marked them in red in the paper. We have tried our best to revise the manuscript based on the comments. Enclosed is the revised version, and we hope to submit it to you. Sincerely, Xiao TangReviewer 3 Report
An attempt was made by the authors to answer some of the questions previously raised, and make some corrections and additions. But at two points I have to mention the following again:
In brief, I refer to the graphs of Figures 6 and 13.
I did not ask for the definition of ring down count (RDC). It is known to those who deal with AE technique. I simply suggested that it is better to calculate the rate of AE RDC as well as AE hits, which is the most common and provides clearer information regarding the AE activity.
It is not clear from the paper that this was followed by the authors. My opinion is that the Figures show the number of RDCs and hits and not the rate.
For the RDC rate and hits authors should consult the following papers:
- A Quantitative Strain Energy Indicator for Predicting the Failure of Laboratory‑Scale Rock Samples: Application to Shale Rock, Rock Mechanics and Rock Engineering (2018) 51:2689–2707.
- Investigation of the dependence of deformation mechanisms on solute content in polycrystalline Mg–Al magnesium alloys by neutron diffraction and acoustic emission, Journal of Alloys and Compounds 642 (2015) 185–191.
- An Alternative Approach for Representing the Data Provided by the Acoustic Emission Technique, Rock Mechanics and Rock Engineering (2018) 51:2433–2438.
Note: One way is to count RDCs and hits in a time window that can be determined by the total number of hits. It can be 1 sec whenever we refer to AE RDC or hits per sec, or for example 10 sec whenever we refer to AE RDC or hits rate.
Reviewer 4 Report
The paper has been modified based on the reviewers' comments. It can be accepted in the present form.
Author Response
Dear editor Victor Ibanescu and reviewers: On behalf of my co-author, thank you very much for providing us with the opportunity to revise the manuscript, and thank the editors and reviewers for their positive and constructive contributions to our manuscript entitled "Tensile Damage Research on Wind Turbine Towers" Comments and suggestions. Material Q345" based on acoustic emission method. (ID: materials-1168197). We have carefully studied the reviewer’s comments, revised them, and marked them in red in the paper. We have tried our best to revise the manuscript based on the comments. Enclosed is the revised version, and we hope to submit it to you. Sincerely, Xiao Tang